behaviour/cognition/physiology

cooperation, loose-string paradigm, *Macaca fascicularis*, salivary cortisol, glucocorticoid hormone, social bonds

**Author for correspondence:**
Martina Stocker
e-mail: martina.stocker@univie.ac.at

# Cooperation with closely bonded individuals reduces cortisol levels in long-tailed macaques

Martina Stocker[1,2], Matthias-Claudio Loretto[1,3,4], Elisabeth H. M. Sterck[2,5], Thomas Bugnyar[1] and Jorg J. M. Massen[1,5]

[1]Department of Cognitive Biology, University of Vienna, Vienna, Austria
[2]Animal Science Department, Biomedical Primate Research Centre, Rijswijk, The Netherlands
[3]Department of Migration, Max Planck Institute of Animal Behavior, Radolfzell, Germany
[4]Department of Biology, University of Konstanz, Konstanz, Germany
[5]Animal Ecology, Utrecht University, Utrecht, The Netherlands

MS, 0000-0002-4736-0009; M-CL, 0000-0002-1940-3470; EHMS, 0000-0003-1101-6027; TB, 0000-0002-6072-9667; JJMM, 0000-0002-1630-9606

Many animal species cooperate with conspecifics in various social contexts. While ultimate causes of cooperation are being studied extensively, its proximate causes, particularly endocrine mechanisms, have received comparatively little attention. Here, we present a study investigating the link between the hormone cortisol, cooperation and social bonds in long-tailed macaques (*Macaca fascicularis*). We tested 14 macaques in a dyadic cooperation task (loose-string paradigm), each with two partners of different social bond strength and measured their salivary cortisol before and after the task. We found no strong link between the macaques' cortisol level before the task and subsequent cooperative success. By contrast, we did find that the act of cooperating in itself led to a subsequent decrease in cortisol levels, but only when cooperating with closely bonded individuals. Two control conditions showed that this effect was not due to the mere presence of such an individual or the pulling task itself. Consequently, our study shows an intricate way in which the hypothalamic–pituitary–adrenal axis is involved in cooperation. Future studies should reveal whether and how our findings are driven by the anxiolytic effect of oxytocin, which has been associated with social bonding.

# 1. Introduction

Cooperation can be described as the voluntary acting together of two or more individuals, (potentially) leading to an end situation that benefits at least one of the participants [1]. A multitude of animal species, as diverse as chimpanzees, *Pan troglodytes* [2]; lions, *Panthera leo* [3]; ravens, *Corvus corax* [4]; cichlid fish, e.g. *Astatotilapia burtoni* [5] or eusocial paper wasps, *Polistes dominula* [6] cooperate with conspecifics in various social contexts such as cooperative territory defence [3,5] and hunting [7,8]. Apart from the evolutionary function of cooperation (i.e. fitness benefits [9]), during the past decades, the cognitive mechanisms underlying cooperative behaviour have received specific attention in behavioural and psychological research. Cooperating individuals may require a suite of cognitive traits, allowing them, for example, to recognize and assess their partners, to tolerate spatial proximity of others, to coordinate their actions with others to achieve a particular aim, and, at times, to accept lower immediate pay-offs [10]. These cognitive abilities are tightly linked to physiological processes, including hormonal mechanisms modulating cooperative behaviour [10–12] that have, however, received considerably less attention.

Glucocorticoid hormones help individuals to cope physiologically with fluctuating environmental factors [13]. Changes in glucocorticoid levels can lead to changes in cooperative behaviour [14,15], yet so far there is no clear picture of this relationship. In some cases, e.g. the cleaner wrasse, *Labroides dimidiatus*, the effect of glucocorticoids on cooperation was shown to be condition dependent [15]. Individuals that were injected with cortisol were less cooperative (i.e. biting instead of cleaning) towards potentially dangerous large clients, whereas they were more cooperative towards relatively harmless small clients, suggesting that cortisol mediates cooperative flexibility [15]. Altogether, however, still relatively little is known about the link between cooperation and glucocorticoids.

The identity of the cooperation partner represents another very important component when it comes to the performance of cooperative actions. One of the most influential features is the social relationship two individuals share with each other. Particularly close social bonds, reflected by high grooming rates and the individuals often being in close proximity [16–19], have been shown to promote cooperation in a broad range of species (e.g. kea, *Nestor notabilis* [17]; wolves, *Canis lupus* [20]; chimpanzees [21]; Barbary macaques, *Macaca sylvanus* [22–24]). Chimpanzees, for example, are more likely to share food with closely bonded conspecifics [21], and in male Barbary macaques, close social bonds facilitate cooperation in coalitions against other males [22,24].

Besides positively affecting cooperation, close social bonds can also buffer hypothalamic–pituitary–adrenal (HPA) axis activity, which means they can lower glucocorticoid levels [25]. This has also been found in several primate species, such as (female) yellow baboons, *Papio cynocephalus* [26,27]; Barbary macaques [28]; chimpanzees [29] and also humans [30]. While many studies have shown that this applies to stressful situations in which responses to stressors are alleviated by interactions with and also the mere presence of closely bonded individuals (social buffering hypothesis [25,28]), there is also evidence that this is a general effect that applies to different contexts throughout daily life, independent of immediate stressors (main effects hypothesis [25,29]).

Here, we investigate the relation between cooperation, social bonds and endogenous glucocorticoid levels in long-tailed macaques, *Macaca fascicularis*. This macaque species is relatively despotic, lives in multi-male multi-female groups with a steep linear hierarchy [31] and is well suited for this investigation for the following reasons: they cooperate through alliances in agonistic contexts [32]; they support each other in exchange for grooming [33] and also they reciprocate grooming for itself [34]. Moreover, in experimental settings, long-tailed macaques can show pro-social behaviour [35,36]. Since they are averse to unequal food distributions [37], they only do so as long as it does not come at a cost [35]. Long-tailed macaques' pro-social choices mainly depend on actor and receiver's rank position [19,36], although there are minor indications that also social bonds may play a role in pro-social choices [19]. In addition, studies on a closely related, but more egalitarian species, the Barbary macaque showed that strong social bonds in fact drive the maintenance of cooperative interactions over time [23,24], and that affiliative interactions are related to lower cortisol levels [38] as well as higher oxytocin levels [39]. Social bonds, like rank [19,22,36], thus, seem to play an important role in macaque cooperation.

Glucocorticoids (or their metabolites) in long-tailed macaques have so far mainly been measured from blood, urine or faeces [40–44]. Saliva sampling, however, provides a non-invasive alternative that allows measuring short-term changes in unbound, hence biologically active cortisol [45]. Due to this advantage, saliva sampling is being applied more and more frequently in non-human primates [46–52] and is also the method of choice in this study.

Here, we aimed to gain a better understanding of the potential endocrine and social mechanisms underlying cooperation. In particular, we want to address three questions: *Q1*: Is an individual's cooperative behaviour in an experimental task modulated by its cortisol level before the task and/or by the social bond with its cooperation partner? *Q2*: Are subsequent changes in the individual's cortisol

level influenced by the performance in the cooperation task; i.e. do cortisol levels change due to cooperation, and/or due to other social factors, such as the social bond between participants? Q3: Does the test setting itself, or only the cooperative aspect of it affect subsequent cortisol levels?

We tested 14 long-tailed macaques in a dyadic loose-string task [53]—a task that has already been used to test cooperative tendencies in primates [2,23,53,54], other mammals [55,56] and birds [4,17,18,57–59], and which requires two individuals to collaborate (i.e. simultaneously pulling on the two ends of a rope) to access a sharable reward. Before and after the task, we measured their salivary cortisol levels. Individuals were tested twice with group members of varying social bond strength, ranging from closely bonded conspecifics to rather neutral ones. We predicted cortisol before the task to be negatively and social bond to be positively related to cooperative success, and that these effects would be independent of subject's rank (see [36]) (P1). In addition, we expected that more successful individuals would experience a decrease in cortisol during the task, yet that a more pronounced drop in cortisol would occur after cooperation with closely bonded group members, and that these effects would be independent of possible effects of sex and dominance rank on cortisol levels (see [60,61]) (P2). Finally, we predicted that only cooperation and not the mere presence of a conspecific or the pulling task itself reduces cortisol (P3).

# 2. Material and methods

## 2.1. Subjects and housing

We tested 14 long-tailed macaques (five males, 3–5 years; nine females, 4–22 years) living in a captive social group of 29 individuals at the Biomedical Primate Research Centre (BPRC) in Rijswijk, The Netherlands. Test subjects were above 2 years of age and passed the training criteria (see electronic supplementary material, 1: *Training procedure*). The group was kept in an enclosure consisting of freely accessible indoor (72 m$^2$, 2.85 m high, light/dark cycle 12 h) and outdoor compartments (208 m$^2$, 3.1 m high). Plenty of structures, e.g. platforms, tyres and ladders, served as environmental enrichment. The animals were fed twice a day with monkey chow pellets (Sniff; 9.00), and either fresh fruit, vegetables or bread (15.00). Water was available ad libitum.

## 2.2. Social measures

### 2.2.1. Dominance hierarchy

The dominance hierarchy within the group was determined by Landau's linearity index corrected for unknown relationships ($h'$) using MatMan 1.1 [62]. The calculation of the index was based on 132 occurrences of submissive behaviour, i.e. bare teeth displays and being displaced, collected on an all occurrence basis (9.00–17.00, indoors and outdoors) throughout the study period ($h' = 0.338$, $n = 18$, without individuals below 2 years of age, $p = 0.02$, 54.25% unknown relationships).

### 2.2.2. Social bond

To determine the social bond between individuals, we used the frequency of them being observed in close proximity (within one arm's length, including grooming incidents) during scan samplings. Since these relationships changed slightly during a testing break (1.5 months: 17 August to 2 October 2016), we calculated the scores for the testing period before and after the break using scans taken during the respective period plus one month before (period 1: 61 scans from 29 days; period 2: 40 scans from 27 days; maximum 3 scans/day, at least 45 min in between). As the relationship of two partners might be different for each partner (whereas one might spend much more time with others, for the other the partner might be the one he/she shows most of the proximity occurrences with), we calculated the individuals' proportions of proximity with each other group member (older than 2 years of age), based on the total proximity frequency of the respective individual, rather than on a group level. Proportions ranged from 0 to 0.39 (39% of this individual's total proximity occurrences took place with this partner). Test partners (see below) were matched in a way that each individual was tested with one conspecific with a high social bond proportion and one with a low proportion. We avoided testing individuals that showed high levels of aggression towards each other. Due to this logistically challenging matching procedure, we did not always pick the individual with the highest social bond proportion as a strongly bonded partner. Out of 16 tested dyads, six had a maternal kin relationship with $r \geq 0.25$ (one sibling dyad, two mother–daughter dyads, three aunt/uncle–niece/nephew dyads).

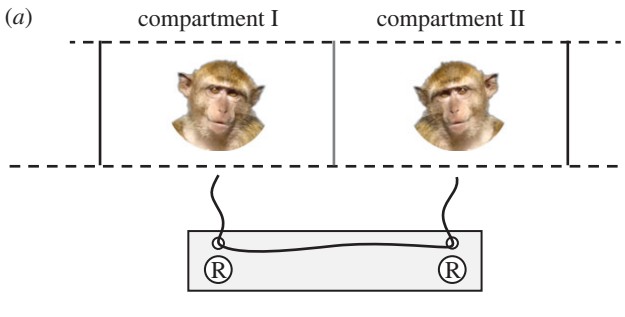

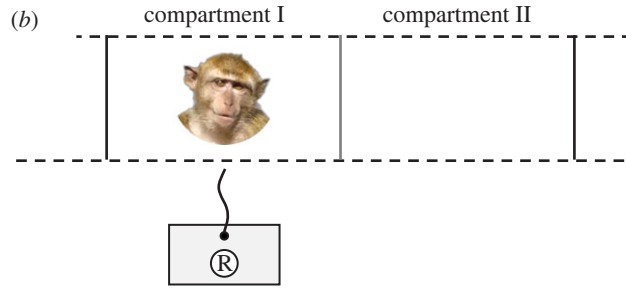

**Figure 1.** Test settings: (*a*) Dyadic setting (loose-string paradigm) and (*b*) individual setting with fixed rope. The grey line between the two compartments represents a transparent slide, the grey rectangle the sliding platform and R the reward (peanuts).

## 2.3. Experimental set-up

Subjects were tested inside a corridor of their indoor enclosure, which, for the experiment, was divided into two smaller compartments (I and II, each $110 \times 75 \times 80$ cm) with a transparent slide in the middle and opaque slides on both ends (figure 1). During testing subjects were kept separately in the two compartments to avoid physical aggression and grooming interactions which might affect cortisol levels [63]. Depending on the condition (see §2.5 *Conditions*), either a dyadic or an individual test apparatus was placed in front of the corridor.

### 2.3.1. Dyadic test setting

The dyadic test apparatus was based on the loose-string paradigm [53] (figure 1*a*; for dimensions, see electronic supplementary material, 1) and was used in the cooperation condition (see §2.5 *Conditions*). In this set-up, each macaque could reach just one end of a rope that was threaded through loops attached to a sliding platform. Only when the subjects pulled the ends simultaneously, the platform moved within reach and the subjects could get the rewards (peanuts) that were placed on this platform. If only one individual pulled, the rope got disentangled and the dyad lost its chance to cooperate during that trial.

### 2.3.2. Individual test setting

The individual test apparatus (figure 1*b*; for dimensions see electronic supplementary material, 1) was used in the non-social control condition (see §2.5 *Conditions*). It was presented in front of compartment I and differed from the dyadic apparatus in that the rope was not loose but fixed on the (smaller) sliding board. In this task, one macaque alone had to pull the rope to bring the sliding board including a reward that was placed in the cup within reach.

## 2.4. Training

All subjects were familiar with coming into the corridor and being separated (physically, but not acoustically or visually) from their group. For the purpose of this experiment, we first trained the animals on the individual string-pulling task (figure 1*b*) and subsequently on the dyadic loose-string task (figure 1*a*; for training procedure see electronic supplementary material, 1).

**Table 1.** Test conditions including short description and purpose.

| condition | description | effect of… |
|---|---|---|
| cooperation | loose-string task with a partner | … cooperation? |
| social control | same partner, but no task | … conspecific's presence? |
| non-social control | no partner, individual string-pulling task | … string-pulling/activity? |

## 2.5. Conditions

The macaques were tested in three different conditions in a within-subject design (overview table 1). To examine the individuals' cooperative success and its relation to their hormone levels, we applied (i) the loose-string paradigm; i.e. *cooperation condition* (figure 1*a*). Since changes in hormone levels could, however, not only arise from cooperation but due to other reasons, we implemented two controls (ii) a *social control condition*, controlling for the possible effect of a partner's presence and (iii) a *non-social control condition* to determine the effect of the task, hence the pulling (activity) itself. In the social control, the same dyads had to run through the same procedure as in the cooperation condition but were not presented with a task. In the non-social control, subjects were alone in the corridor and confronted with the individual test setting (figure 1*b*).

To test the effect the social bond with the cooperation partner may have on either the cooperation itself or on the mediating role of hormones, we tested subjects in the cooperation and social control condition with a closely bonded group member and with a neutral member (see §2.2.2 *Social measures–Social bond*). Therefore, individuals were tested five times; twice in the cooperation test, twice in the social control and once in the non-social control.

Social conditions (i.e. cooperation and social control) as well as the social bond with the partner were counterbalanced over the individuals (see electronic supplementary material, 1: *Counterbalanced testing*); the non-social control was conducted after an individual was tested with one partner in both social conditions. Since for two subjects we lacked partners with appropriate social bond patterns who had not been tested yet, two already tested individuals had to be used as partners and thus were tested one additional time in both social control and cooperation test (for dyads see electronic supplementary material, 1 and table S1). The tests were conducted between 30 June and 11 November 2016.

## 2.6. Experimental procedure

At the beginning of each test session, the respective individual or dyad was led into the test compartments and separated from the rest of the group. Females with an infant (4–9 months old) were allowed to bring it to the test compartments ($n = 4$). After 5 min of habituation they were confronted with one of the three conditions. Each session had 15 trials (of 35 s each) with an inter-trial interval of 40 s, equalling exactly 10 min. In the cooperation and non-social control condition, an experimenter reset the apparatus after each trial (retracted platform and rope, baited cups, reoffered rope). In these conditions, the monkeys thus got the opportunity to, respectively, cooperate to sequentially get 15 rewards (peanut halves) and to pull the tray on their own to get 15 rewards. Since we assumed that the subjects would not succeed in all 15 trials and to keep conditions as standardized as possible, we chose to simulate 12 successful cooperation trials in the social control condition. Therefore, in the social control condition, the experimenter gave each individual one reward per trial (no task), except for trial 5, 10 and 15, equalling 12 successful cooperation trials. The whole study was conducted by the same person who was always present during the sessions. After the 10 min test sessions, we took two saliva samples: the first one 5 min and the second one 10 min after the test (electronic supplementary material, 1 and figure S1), always starting with the same individual. Prior to the experiments, we performed a biological validation of salivary cortisol in long-tailed macaques (see electronic supplementary material, 2), which revealed that salivary cortisol has a time lag of approximately 15 min. Therefore, sample 1 should reflect what happened at the beginning of the test (cortisol before), and sample 2 represents the test situation. Following the last saliva sample, the macaques were released back into their group. To limit the variation in cortisol levels due to circadian rhythm (for rhesus macaques, *Macaca mulatta*, see [46]), all test sessions were conducted between 13.00 and 15.00. Individuals were never tested twice per day.

**Table 2.** Overview of the full models corresponding to the research questions Q1–3, including the data used (condition; sample size (number of observations)), dependent and independent variables. All models include individual ID as a random factor. Model Q1 also includes partner ID as random factor. Condition: CO, cooperation; SC, social control; NS, non-social control; Δ cortisol, change in cortisol during the condition; coop., cooperative; soc., social; m. kin, maternal kin.

|      | condition  | n (obs.) | dependent v.   | independent variables |           |      |     |        |        |
| ---- | ---------- | -------- | -------------- | --------------------- | --------- | ---- | --- | ------ | ------ |
| **Q1**   | CO         | 14 (29)  | coop. success  | cortisol before       | soc. bond | rank |     |        |        |
| **Q2 a** | CO         | 13 (27)  | Δ cortisol     | coop. success         | soc. bond | rank | sex | infant | m. kin |
| **Q2 b** | SC         | 14 (27)  | Δ cortisol     |                       | soc. bond | rank | sex | infant | m. kin |
| **Q3**   | CO, SC, NS | 14 (66)  | Δ cortisol     | condition             |           | rank | sex | infant |        |

## 2.7. Salivary cortisol

### 2.7.1. Sample collection procedure

Using positive reinforcement, the subjects were trained to give saliva samples by chewing on one end of a Salimetrics Children Oral Swab (previously validated for common marmosets, *Callithrix jacchus* [47]), which was held on the other end by the experimenter (wearing gloves). For each sample the animal was rewarded with a small piece (∼ 1/8) of peanut. The swabs were collected in 2 ml micro-tubes and, after at most 90 min, frozen at −20°C until analysis (for general information on freezing and storage see [50]).

### 2.7.2. Hormone analysis

The collected samples were thawed, centrifuged at 10 000 r.p.m. for 10 min (Eppendorf Centrifuge 5424, rotor FA-45-24-11), diluted 1 : 5 with diluent buffer (IBL International: KLZZ731; appropriate ratio determined beforehand by serial dilution of pooled saliva) and analysed with a commercial enzyme immunoassay kit (IBL International: Cortisol Saliva ELISA RE52611) following the instructions of the manufacturer, where also cross reactivities can be found [64]. For validation see electronic supplementary material, 2. The intra-assay coefficient of variance (CV) was 7.13% and the inter-assay CV 9.99%. Saliva samples were analysed in duplicates; however, in some cases, the macaques did not provide sufficient saliva for cortisol analysis (sample size, see table 2), or the volume only sufficed for single analysis. These values were nevertheless included in the statistical analysis, since only one out of 165 duplicates (= 0.6%) had a CV above 10%, indicating that the assay was highly reliable. The analysis was conducted at the Biomedical Primate Research Centre within two months after sample collection.

Long-tailed macaques' salivary cortisol levels ranged from 1.45 to 43.55 ng ml$^{-1}$, with a mean of 5.67 ng ml$^{-1}$ ± 4.35 s.d. Three samples were excluded from further analysis, because they exceeded the individual's mean + 2.5 s.d. This resulted in a total mean of 5.27 ng ml$^{-1}$ ± 2.71 s.d. and a maximal value of 16.85 ng ml$^{-1}$.

## 2.8. Statistical analyses

All statistical analyses were conducted with R version 3.3.3 [65]. Linear mixed effect models were calculated with the package *lme4* [66], except for the full model associated with Q1, which was a generalized linear mixed model (GLMM) with β-distribution constructed with the package *glmmADMB* [67]. In this case, the response variable cooperative success (which is limited to a maximum of 15 trials) was better represented by a β-distribution than by a Poisson (open-end distribution) or normal distribution and was therefore given as proportion instead of number of successful cooperation trials per session. All full models, including data used, response variable and independent variables are listed in table 2 (for model summaries see electronic supplementary material, table S2). Due to the individual nature of cortisol levels and the fact that they are included in all models, we based our analyses on individuals and not dyads. All full models included the individual's identity as random factor. Model Q1 additionally included the partner's identity to account for the direct dependency of cooperative success of the two partners. Changes in cortisol levels (Δ cortisol) were calculated by subtracting the value of the first saliva sample from that of the second one. Positive values of Δ cortisol, therefore, represent an increase of cortisol. Since some of the participating females had an infant, which might have affected their cortisol levels, the presence of an infant was included as an independent variable in models with

**Table 3.** Model-averaged coefficients (full averaging from top-ranked models with standardized parameters) with adjusted standard errors, lower and upper confidence intervals and relative importance. For Q1 and Q2b model averaging was not conducted because the top-ranked models included the null model. s.e., standard error; CI, confidence interval. Relative importance: italic, highly important.

| model and response variable | parameter (level) | estimate | adjusted s.e. | lower CI (2.5%) | upper CI (97.5%) | relative importance |
|---|---|---|---|---|---|---|
| **Q2a** change in cortisol during cooperation | intercept | −0.36 | 0.31 | −0.96 | 0.24 | |
| | rank | 1.61 | 0.62 | 0.39 | 2.83 | *1.00* |
| | social bond | −0.89 | 0.60 | −2.02 | −0.15 | *0.82* |
| | sex (male) | 0.35 | 0.58 | −0.16 | 2.09 | 0.37 |
| **Q3** changes in cortisol during cooperation, social control and non-social control | intercept | −0.17 | 0.18 | −0.52 | 0.17 | |
| | rank | 1.07 | 0.37 | 0.36 | 1.79 | *1.00* |
| | sex (male) | 0.62 | 0.48 | 0.06 | 1.55 | *0.77* |
| | infant (yes) | 0.20 | 0.38 | −0.26 | 1.47 | 0.33 |

cortisol as response variable, as were the individuals' sex, dominance rank and kinship [60]. We inspected diagnostic plots of the model residuals for homogeneity of variance, violation of normality assumptions or other departure from model assumptions and calculated the variance inflation factors (maximum VIF across all models = 1.85), which indicated that the independent variables showed no significant collinearity.

We followed an information-theoretic approach [68], as this is suitable when the number of subjects is limited, and the model should incorporate potentially confounding variables [69], such as the presence of an infant. We created a set of candidate models with all possible combinations of independent variables from the respective standardized full model. We ranked these models according to their AICc and selected the subset of models with ΔAICc ≤ 2 with respect to the best model (electronic supplementary material, 1 and table S3), as this gives substantial support that these models show an equally good fit. With the *MuMIn* package [70], the models of the respective subset were averaged to create model-averaged coefficients [68], which were ranked based on their relative importance (table 3). The relative importance is the crucial value for further discussion of the independent variables (for details on this method see [71]). If this subset included the null model (intercept only model), we do not report these results in detail since it supports the null hypothesis, i.e. none of the models explains the variance within the response variable better than the null model [68].

## 3. Results

Overall cooperation success rate ranged from 6 to 15 trials (40–100%) per session, with a median of 12 trials. When considering this cooperative success, the subset of the best models from all candidate models contained the null model (electronic supplementary material, table S3), indicating that neither cortisol levels before the cooperation task, nor social bond or rank predict how successful individuals cooperated with each other (P1).

Changes in cortisol levels during the cooperation condition were also not affected by the individual's performance in the task (i.e. cooperative success; P2a). An individual's dominance rank and the social bond it shared with the cooperation partner, however, did seem to have a strong effect on cortisol changes while cooperating (table 3). Whereas higher ranking individuals and individuals with a stronger social bond with their cooperation partner were more likely to experience a drop in cortisol, lower ranking individuals as well as weakly bonded partners were more likely to show stable cortisol levels or a slight increase (figure 2).

With regard to the social control, the subset of the best models from all candidate models included the null model (electronic supplementary material, table S3). Therefore, the individuals' social bond did not affect changes in cortisol in the social control (figure 2). This effect, hence, only occurs when individuals are actually cooperating and not while just sitting next to each other (i.e. during social control).

Whereas cooperation with closely bonded individuals led to a cortisol decrease, changes in cortisol were not affected by cooperation in general (irrespective of the bond with the partner), the mere presence of a conspecific, or the pulling task itself (P3). This was indicated by the fact that none of the

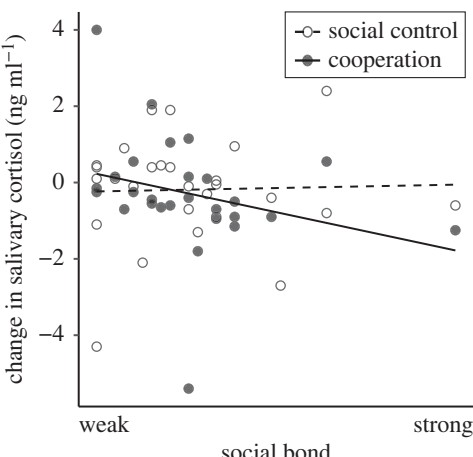

**Figure 2.** Relationship of changes in salivary cortisol levels *during the social conditions* (cooperation, social control; split for illustration) and social bond with the cooperation partner. Since individuals were tested with different partners, there are multiple points for each individual. Cooperation: filled circles, solid trend-line. Social control: open circles, dashed line.

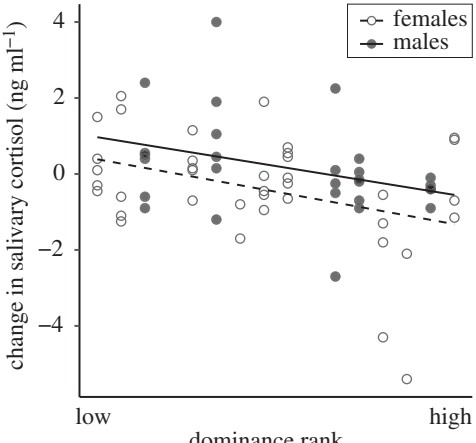

**Figure 3.** Relationship of changes in salivary cortisol levels *across all conditions* and individual dominance rank in males (filled circles, solid trend-line) and females (open circles, dashed line). As the figure presents data from all conditions, there are multiple data points for each individual.

models in the subset used for the model averaging contained the factor condition (electronic supplementary material, 1 and table S3). Instead, model-averaged results show that across all conditions (i.e. cooperation as well as both social and non-social control) dominance rank and sex were the most important predictors of those changes (table 3; figure 3).

## 4. Discussion

We investigated the possible link between cooperation and salivary cortisol in 14 long-tailed macaques. In contrast with our hypothesis, we found no evidence that cortisol levels at the beginning of a cooperation task or the strength of social bond with the partner affect an individual's subsequent cooperative behaviour. Also contrasting with our expectations, cooperative success did not affect subsequent cortisol levels. Hence, similar to recent findings in red-tailed monkeys, *Cercopithecus ascanius* [72], the endocrine responses measured were not related to the *outcome* of the social interaction. Instead, two other factors that rather reflect the nature of the interaction, seemed to influence cortisol responsivity; i.e. dominance rank and social bond. First, during cooperation sessions salivary cortisol levels of higher ranking individuals dropped, whereas for lower ranking individuals, it either stayed the same or even increased. Second, cortisol levels decreased more in animals that worked together with closely bonded individuals. The latter effect seemed specifically based on working together since we did not find a similar pattern in the social control.

Among the long-tailed macaques in this study, there was no evidence for social bonds to influence cooperative behaviour. This contrasts with studies on, for example, chimpanzees [21] or the closely related, yet more egalitarian Barbary macaques [22–24], that did find a positive effect of social closeness on cooperation. It should be noted, however, that, unlike in these studies, that were conducted in the wild, in our experiment the subjects did not have free partner-choice, which is a very important factor in cooperative decision-making [73]. Restricting such partner-choice has been suggested to obscure the effects of social parameters [74]. Moreover, the macaques in our study performed generally very well in the cooperation task (mean 75.6%, median 12 successful trials (indeed equalling the number of rewarded trials in the social control condition) and a range of 6 to 15 successful trials) leaving little variation to relate to social bond strength.

Irrespective of cooperative success, social bond seemed to have a large effect on cortisol changes during the cooperation test, with closer social bonds leading to a larger decrease in cortisol levels. This effect was not found in the social control, suggesting that it is actually the cooperative action with a closely bonded individual and not just its presence that is central to reducing cortisol. In wild chimpanzees, close social bonds also have a downregulating effect on the HPA axis, though not just during cooperation but generally throughout their daily life [29] (supporting the main effect hypothesis [25]). Though in Barbary macaques, affiliative interactions (between males) have been reported to reduce cortisol levels [38], it has been shown that the stress buffering effect of strong social bonds depends on social and environmental factors and is therefore not always the same [28]. The here-demonstrated anxiolytic effect of cooperative interactions with closely bonded individuals could be the underlying proximate cause of the effect of close social bonds on the maintenance of cooperation found in macaques [23,24]. One potential mechanism leading to a decrease in cortisol during cooperation of closely bonded individuals could involve oxytocin. Oxytocin plays a major role in facilitating the formation and maintenance of social bonds [39,75–77] and mediates cooperative behaviour [11] (equivalent mesotocin in birds [78]). Moreover, oxytocin possesses anxiolytic properties [30,79,80] (for non-human primates see [81,82]). Accordingly, we would expect oxytocin levels after socio-positive interactions with closely bonded conspecifics to be particularly high and therefore the anxiolytic effect, eventually leading to decreased cortisol levels, to be strong. In wild chimpanzees, cooperative behaviours like grooming and food sharing were linked to higher peripheral urinary oxytocin levels compared with after social feeding situations that do not involve cooperation [83]. However, unlike the effect of cooperation on cortisol levels in the present study and grooming on oxytocin in chimpanzees [83,84], the effect of food sharing on oxytocin was independent of the social bond shared by the respective individuals [21,83]. Also in the context of inter-group conflicts, which, too, require group members to cooperate, chimpanzees were found to have higher oxytocin levels compared with controls [85], though the release of oxytocin was independent of cortisol activity [86]. Hence, whether the oxytocinergic system affects HPA axis activity during cooperation appears to depend on the specific type of cooperative action. To conclude, in our study, closely bonded individuals did not cooperate more, but their cooperative interaction might have led to an increase in oxytocin, which in turn reduced cortisol levels. If by these means frequent cooperation counteracts chronically elevated glucocorticoid levels, it might promote the maintenance of cooperation over time [23,24] and improve the individuals' health status [87] (but see [88]). This hypothesis, however, still remains to be tested in long-tailed macaques.

Dominance rank was found to predict changes in cortisol across all conditions (figure 3), i.e. in high-ranking individuals the salivary cortisol level was more likely to drop, whereas in low-ranking individuals, it rather increased. This is in line with other studies on long-tailed macaques in which (experimentally manipulated) cortisol changes were rank dependent. Shively *et al.* [40,89] reported a less intense cortisol increase in response to a cortisol-enhancing factor (adrenocorticotropic hormone injection) as well as a stronger decrease in cortisol levels in response to a cortisol-suppressing factor (dexamethasone) in high-ranking females compared with low-ranking ones. The latter was also found in rhesus macaques [90]. Accordingly, our finding might reflect a general pattern in macaques. One potential limitation of our study is that not all dominance relationships were known (see §2.2.1 *Social measures—Dominance hierarchy*) as the determination of the social characteristics, i.e. social bonds and dominance rank, was based on a limited number of observations. Though this might be regarded as a major weakness, we do consider our rank calculation to be a reliable estimate of an individual's general positioning in the dominance hierarchy, indicating whether it is high or low ranking. Moreover, we used rank instead of rank differences between individuals in the statistical analyses because potential inaccuracies in the calculated ranking have major consequences for rank differences between individuals, though should be less impactful when it comes to assessing if an individual is high or low ranking. Furthermore, the fact that the link between rank and cortisol changes found in this study is in line with previous findings in macaques is supporting the validity of our rank data.

The test setting, i.e. the three conditions as such, did not influence subsequent cortisol levels (Q3), yet we did find an overall effect of sex. Regardless of the condition, males in our study were higher in the cortisol reactivity spectrum than females of similar rank, i.e. low-ranking males showed a more pronounced cortisol increase than low-ranking females and high-ranking females a stronger decrease than high-ranking males (figure 3). Similar sex differences in salivary cortisol responses to acute psychological stressors were found in humans [91,92]. These differences could result from the type of social context that causes the response, as different contexts might have different effects in males than in females [93] and might engage different neuroendocrine pathways [91].

# 5. Conclusion

In summary, we investigated the relationship between glucocorticoids and cooperative behaviour in an experimental task in long-tailed macaques, while taking into account the individuals' social bonds. We found no evidence for cortisol or close social bonds to affect the macaques' cooperative performance (similar to [19]). Strong bonds did, however, lead to decreasing glucocorticoid levels during cooperative interactions, irrespective of the performance. We propose that this could be mediated by the anxiolytic effect of oxytocin, which probably is upregulated during cooperative interactions with a closely bonded individual, but not necessarily due to the mere presence of such an individual. Finally, we found that the macaques' cortisol responsivity was affected by their dominance rank and sex. This effect was, however, independent of the conditions and not specific to cooperation. Overall, these findings contribute to our understanding of the mediating role of cortisol in cooperation. However, to get a better understanding of the physiological processes involved in cooperation, we encourage future studies to investigate multiple biomarkers at once [11,12,44,72,87].

Ethics. The study complied with the Association for the Study of Animal Behaviour Behaviour/Animal Behaviour Society Guidelines for the Use of Animals in Research (Animal Behaviour, 2018, 135, I-X) and was carried out after approval of the BPRC's Animal Welfare Committee (IvD 002a) and in accordance with the Dutch legal requirements. The macaques were never water or food deprived and training/testing was based on their voluntary participation.
Data accessibility. All data and R scripts are archived in the Dryad Digital Repository: https://dx.doi.org/10.5061/dryad.g6kh12j [94].
Authors' contributions. M.S. and J.M. conceived of the study and designed it; M.S. carried out animal training, data collection, hormone and data analysis and drafted the manuscript; M.S. and M.L. carried out the statistical analyses; E.S. and T.B. contributed to the interpretation of the findings; J.M. coordinated the study and helped draft the manuscript. All authors gave final approval for publication.
Competing interests. We have no competing interests.
Funding. M.S. and J.M. were supported by the FWF (Austrian Science Fund, P 26806 to J.M.) and the IPS (International Primatological Society, Research Grant to J.M. and M.S.); M.L. was supported by the FWF (Austrian Science Fund, P 29705) and by the European Union's Horizon 2020 research and innovation programme under the Marie Sklodowska-Curie grant agreement no. 798091.
Acknowledgements. We thank Annet Louwerse and Jan Langermans for allowing us to conduct the study at the BPRC, Caroline Olesen, Lisette van den Berg, Saskia Visser-van Soest and Willem van der Spek for supporting us in the work with the macaques, Daniella Mortier, Linda Hofman, Elisabeth Pschernig and Ruth Sonnweber for their help/advice concerning the hormone analyses, and Tjeerd Fluitsma for building the test apparatuses. Finally, we would like to thank Loeske Kruuk, Sarah Brosnan and the anonymous reviewers for their constructive comments on earlier versions of this manuscript.

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
