## [Reviewer comments · Royal Society Open Science]

Review History

RSOS-191056.R0 (Original submission)

Review form: Reviewer 1 (Cedric Girard-Buttoz)

Is the manuscript scientifically sound in its present form?

No

Are the interpretations and conclusions justified by the results?

No

Is the language acceptable?

Yes

Do you have any ethical concerns with this paper?

No

Have you any concerns about statistical analyses in this paper?

Yes

Recommendation?

Major revision is needed (please make suggestions in comments)

Comments to the Author(s)

In this interesting study the authors report one rare experimental test of the effect of collaboration on cortisol excretion. The study is ambitious and appears well controlled and the results are clear cut. I am however not sure about certain of the analysis and the methods still lack certain level of details (despite the revisions). The author should also clearly acknowledge that their definition of cooperation is not universal and that what they describe here is more commonly termed collaboration (acting together to achieve a common goal that one cannot achieve alone) whereas cooperation in the broadest sense (in the Darwinian sense) is defined as a service provided by an individual A to another individual B at a cost for A and which benefits B. In this sense behaviors such as grooming or alarm calling have been frequently given as example of cooperative actions. By taking a broader definition of cooperation the authors could incorporate the literature on physiological consequences of grooming interaction for example on cortisol excretion (and on other hormones such as oxytocin) into their introduction and broaden the scope of their paper. Even sticking strictly to their definition, the authors omit in their introduction some recent literature on physiological correlates of collective collaborative action such as border patrol and hunting in chimpanzees (e.g. Samuni et al. 2017, 2019).

Introduction:

As mentioned above, the introduction would benefit from incorporating broader literature about the link between cooperation (in the road sense of the term) and mammalian physiology. This would also shed light on the theoretical rational of the predictions which partly appear not really justified by the introduction itself as it stands.

Line 58: replace "of" by "between"

Line 65: replace "since" by "while".

Line 81: please define degree of association clearly in the introduction. It is important to understand your hypothesis. Also you can refer to literature showing that in societies where individuals form social bonds with each other (e.g. macaques, baboons, chimpanzees) social proximity or association in party often reflects social bond strength assessed with more direct/intimate measures such as grooming frequency. That would allow you to present your predictions and discuss your results in a social bond framework. I know that previous reviewers criticized that you used the term social bond, I would clearly state that you use association measures as a proxy for social bond strength and that this proxy is likely to be a good one since in other species of macaques (e.g. barbary or Assemese macaques) it was found that proximity strongly correlates with grooming time and overall social bond strength (see the work from Ostner's and Schulke' research group).

Line 86: The authors should not assume that the reader knows what a loose string task is, especially for a broad audience like royal society open science. They may want to clarify the paradigm in one short sentence and mention that this is a collaborative action where two individuals act together to access a sharable reward.

Line 92-96: Why are the predictions named "Q"? Also the prediction Q2 is not really well introduced by the general introduction and comes a bit out of the blue. Using the literature on oxytocin correlates of grooming (e.g. Crockford et al. 2013) would help drawing prediction on differential physiological effect of cooperation (or social interaction) in bonded and non-bonded individuals.

Methods

The methods are overall clear but lack certain level of details in certain section which makes it hard to grasp what the authors exactly did. I am also not convinced that the way the data were analyzed is the most parsimonious and I would strongly advice to reduce the number of analysis and incorporate interactions into full models to truly test if the effect is condition dependent.

Line 155: you call it in the introduction association index. Now it's "quality of the relationship".

Be consistent or clearly state somewhere that you use association index as a proxy for relationship quality.

Line 157-158: Since it is a key parameter in the analysis it would be wise to explain what the associated group members are and what the others are and to put this section of the methods about how it was calculated before the condition section.

Line 181: Why did individual not receive rewards at trial 5, 10 and 15?

Line 207: Has this kit been biologically validated for use in macaques specifically? If so please cite the appropriate literature. I know that assay are really species specific when it comes to cortisol metabolites in feces (see e.g. Heistermann et al. 2006), what is the kit actually measuring in saliva? Please specify.

Line 218: Why is the unit suddenly jumping from ng/ml to ng/g? Also how was saliva concentration controlled for? (in the same way as urine concentration is controlled for using specific gravity or creatinine).

Line 222: By "Landau's linearity index" do you mean that you used the I&SI method based on matrix of winner loser interactions? Please clarify.

Line 226: why did you remove the breeding males from the hierarchy? Weren't they part of the study? Why were other males kept in the study? What defines a breeding male?

Line 227: the % of unknown relationships is really high and this is a problem even if the authors do not consider rank differences but only rank of the participant itself... Furthermore most of the discussion about the effect of rank revolved around the idea that lower ranking individual than their partners do A and higher ranking do B whereas this is not what is tested in the models. I am really surprised that the hierarchy was not clearer since long-tailed macaques have really clear linear matrilineally based hierarchies in females.

Line 231: The authors should explain why they did not consider grooming in their index.

Line 272: Given the clear hypothesis and predictions you have I do not understand why you use a information theory approach especially for an experimental setup. This form of analysis is more suited to exploration of several parameters potentially affecting the response. E.g. a suit of ecological and anthropogenic factors potentially affecting species density or behavior. In the case of this study the parameters are clearly defined and predictions are drawn from each of them so a cohesive approach building a full model (with interaction between test condition and association index) appears more suited and more parsimonious. If the authors want to keep the analysis as is they should at least explain why they went for this approach over the null hypothesis testing approach.

I noticed that none of your model comprises random slopes. Random slopes are really important to be incorporated into model since the lack of random slopes strongly inflate type 1 error rate (see e.g. Barr et al. 2013). I would strongly advice for the inclusion of random slopes which also do not affect directly the degree of freedom (i.e. complexity) of the model and should not therefore be a limitation due to sample size.

As mentioned above, to truly test if the effect of association index on cortisol reduction is truly occurring only during cooperation (or more so during this context than during the control conditions) the authors need to build a full model comprising the interaction term between test condition and association index. This is feasible even within the AIC model selection framework.

Line 282: you cannot say that you don't use the null hypothesis testing framework and then say that you don't support the null hypothesis.

Results:

The general result section refers to table entirely for the numerical values of the results which I would have incorporated to a larger extent directly in the text to understand what the result mean. Also actual figures like "40% of high ranking individuals did X" would help grasping the effect size of the differences.

Line 292: why is it "conversely" if it is "also not"?

Line 296-298: Fig 2 does not depict the results on the rank effect.

Discussion

In line with my comments on the methods, in the discussion the authors make some claims which are not directly supported by their analysis. Overall I found the discussion relatively shallow

when it could have more impact while broadening the literature survey (which is not a limited factor since the number of citations is not limited in royal society open science).

Line 317: add “with” after “contrasting”

Line 321: There is a whole literature on the effect of dominance rank and social interactions (e.g. social buffering, Young et al. 2014) on cortisol excretion which is ignored in the discussion.

Line 322: If you mention “dominant” this refers to the relationship between 2 individuals. If you mean high ranking individuals this is different since it refers to the entire hierarchy. What I mean is that in the test if you pair alpha and beta females they are both high ranking categorically but the alpha female is the only dominant one of the pair. Please rephrase.

Line 326: to know if the pattern is truly different in the social control and cooperation condition you need an interaction term in your full model. (see comment above). If the interaction term is not within your best models then the effect is not condition specific.

Line 328: you did not define “close social relationships” (see comments above).

Line 332: This is not what the citation [51] found. In their study Wittig et al. show that cortisol is decreased when cooperating, grooming or sitting in proximity to bond partners to the same extent (see the non-significant interaction between relationship quality and event in their “main effect model”. I also wonder why this paper is not mentioned in the introduction of the paper. It would give some scientific rationale to the main predictions of the paper.

Line 346: here cite for grooming in chimpanzees Crockford et al. 2013.

Line 354: This entire paragraph speculate about the different effect for the higher and the lower ranking individual of the pair in the test. Yet this is not tested here, all what is tested is the rank of the participant. As mentioned above two low ranking individuals still have a rank difference and one will be dominant over the other one, the same apply to pairs of high ranking individuals so that overall high and low ranking (to the exception of the alpha and lowest ranking individuals) can each be both dominant and subordinate in the pair depending who they are paired with. The authors should therefore refrain from drawing conclusion on the rank relationship of the pair when they don't test it.

Line 364: it is good to acknowledge the limit of your study but please expand on what it means for your overall conclusions.

Additional comment on the reply to reviewers:

Regarding the choice of a beta model: the author's choice is justified not only by the distribution of the data but also because proportions are by nature bounded between 0 and 1 and a beta distribution model is open ended distribution. Just for that reason beta distribution is better suited. This should be clearly stated in the manuscript.

Bibliography

Barr, D. J., Levy, R., Scheepers, C. & Tily, H. J. 2013. Random effects structure for confirmatory hypothesis testing: Keep it maximal. *Journal of Memory and Language*, 68, 255–278.

Crockford, C., Wittig, R. M., Langergraber, K., Ziegler, T. E., Zuberbuehler, K. & Deschner, T. 2013. Urinary oxytocin and social bonding in related and unrelated wild chimpanzees. *Proceedings of the Royal Society B-Biological Sciences*, 280, 20122765.

Heistermann, M., Palme, R. & Ganswindt, A. 2006. Comparison of different enzyme immunoassays for assessment of adrenocortical activity in primates based on fecal analysis. *American Journal of Primatology*, 68, 257–273.

Samuni, L., Preis, A., Mundry, R., Deschner, T., Crockford, C. & Wittig, R. M. 2017. Oxytocin reactivity during intergroup conflict in wild chimpanzees. *Proceedings of the National Academy of Sciences*, 114, 268–273.

Samuni, L., Preis, A., Deschner, T., Wittig, R. M. & Crockford, C. 2019. Cortisol and oxytocin show independent activity during chimpanzee intergroup conflict. *Psychoneuroendocrinology*, 104, 165–173.

Young, C., Majolo, B., Heistermann, M., Schülke, O. & Ostner, J. 2014. Responses to social and environmental stress are attenuated by strong male bonds in wild macaques. *Proceedings of the National Academy of Sciences*, 111, 18195–18200.

Review form: Reviewer 2

Is the manuscript scientifically sound in its present form?

No

Are the interpretations and conclusions justified by the results?

Yes

Is the language acceptable?

Yes

Do you have any ethical concerns with this paper?

No

Have you any concerns about statistical analyses in this paper?

Yes

Recommendation?

Accept with minor revision (please list in comments)

Comments to the Author(s)

I appreciate the authors' detailed responses to the previous comments. The revisions that were made have significantly strengthened the paper.

The one concern I still have is examining whether kinship or the presence of an infant affected cooperative success (model Q1). Both of these factors have significant biological (kin selection theory) and practical (the physical constraints of pulling in a cooperative task while holding an infant) relevance to cooperative success alone. These factors may have been more important in determining cooperative success than the IVs the authors chose to include in Model Q1 (e.g. cortisol before, social association, rank).

Minor comments:

Line 203: It is not sufficient to report rpms without describing the make and model of the centrifuge and model of the rotor. Generally speaking, the radius of rotors in different centrifuges are different. Alternatively, you can report the g/RCF (gravity/relative centrifugal force) since it is a function of the speed that the centrifuge spins and the radius of the rotor.

ESM 2, 2nd paragraph, 2nd sentence: Should "2,5" be "2.5 min"?

ESM 2, 2nd paragraph, 3rd sentence: "...after collection of the first sample and a to the subject..." Please rephrase.

ESM 2, 2nd paragraph, 4th sentence: Please provide a reference for this.

ESM 2, Reference #4: Use sentence case for the title of the article.

Decision letter (RSOS-191056.R0)

18-Sep-2019

Dear Professor Stocker,

The editors assigned to your paper ("Cooperation with closely associated individuals reduces cortisol levels in long-tailed macaques") have now received comments from reviewers. We would like you to revise your paper in accordance with the referee and Associate Editor suggestions which can be found below (not including confidential reports to the Editor). Please note this decision does not guarantee eventual acceptance.

Please submit a copy of your revised paper before 11-Oct-2019. Please note that the revision deadline will expire at 00.00am on this date. If we do not hear from you within this time then it will be assumed that the paper has been withdrawn. In exceptional circumstances, extensions may be possible if agreed with the Editorial Office in advance. We do not allow multiple rounds of revision so we urge you to make every effort to fully address all of the comments at this stage. If deemed necessary by the Editors, your manuscript will be sent back to one or more of the original reviewers for assessment. If the original reviewers are not available, we may invite new reviewers.

- Data accessibility

<http://datadryad.org/submit?journalID=RSOS&manu=RSOS-191056>

- Competing interests

- Authors' contributions

- Acknowledgements

- Funding statement

Best regards,
Lianne Parkhouse
Royal Society Open Science
openscience@royalsociety.org

on behalf of Dr Mark Walton (Associate Editor) and Kevin Padian (Subject Editor)
openscience@royalsociety.org

Comments to the Author:

Thank you for your efforts at revising your paper. It appears that the reviewers are interested in its novelty but still not convinced of the results and some of the methods. We will allow one further revision, if you choose, but our policy is that if all of the reviewers' concerns are not fully satisfied we cannot consider the manuscript further. I recognize that this is a difficult challenge, and I hope you will answer all of the comments so that your reasons for accepting or demurring are clear.

Reviewers' Comments to Author:

Reviewer: 1

In this interesting study the authors report one rare experimental test of the effect of collaboration on cortisol excretion. The study is ambitious and appears well controlled and the results are clear cut. I am however not sure about certain of the analysis and the methods still lack certain level of details (despite the revisions). The author should also clearly acknowledge that their definition of cooperation is not universal and that what they describe here is more commonly termed collaboration (acting together to achieve a common goal that one cannot achieve alone) whereas cooperation in the broadest sense (in the Darwinian sense) is defined as a service provided by an individual A to another individual B at a cost for A and which benefits B. In this sense behaviors such as grooming or alarm calling have been frequently given as example of cooperative actions. By taking a broader definition of cooperation the authors could incorporate the literature on physiological consequences of grooming interaction for example on cortisol excretion (and on other hormones such as oxytocin) into their introduction and broaden the scope of their paper. Even sticking strictly to their definition, the authors omit in their introduction some recent

literature on physiological correlates of collective collaborative action such as border patrol and hunting in chimpanzees (e.g. Samuni et al. 2017, 2019).

Introduction:

As mentioned above, the introduction would benefit from incorporating broader literature about the link between cooperation (in the road sense of the term) and mammalian physiology. This would also shed light on the theoretical rationale of the predictions which partly appear not really justified by the introduction itself as it stands.

Line 58: replace “of” by “between”

Line 65: replace “since” by “while”.

Line 81: please define degree of association clearly in the introduction. It is important to understand your hypothesis. Also you can refer to literature showing that in societies where individuals form social bonds with each other (e.g. macaques, baboons, chimpanzees) social proximity or association in party often reflects social bond strength assessed with more direct/intimate measures such as grooming frequency. That would allow you to present your predictions and discuss your results in a social bond framework. I know that previous reviewers criticized that you used the term social bond, I would clearly state that you use association measures as a proxy for social bond strength and that this proxy is likely to be a good one since in other species of macaques (e.g. barbary or Assemese macaques) it was found that proximity strongly correlates with grooming time and overall social bond strength (see the work from Ostner's and Schulke' research group).

Line 86: The authors should not assume that the reader knows what a loose string task is, especially for a broad audience like royal society open science. They may want to clarify the paradigm in one short sentence and mention that this is a collaborative action where two individuals act together to access a sharable reward.

Line 92-96: Why are the predictions named “Q”? Also the prediction Q2 is not really well introduced by the general introduction and comes a bit out of the blue. Using the literature on oxytocin correlates of grooming (e.g. Crockford et al. 2013) would help drawing prediction on differential physiological effect of cooperation (or social interaction) in bonded and non-bonded individuals.

Methods

The methods are overall clear but lack certain level of details in certain section which makes it hard to grasp what the authors exactly did. I am also not convinced that the way the data were analyzed is the most parsimonious and I would strongly advice to reduce the number of analysis and incorporate interactions into full models to truly test if the effect is condition dependent.

Line 155: you call it in the introduction association index. Now it's “quality of the relationship”. Be consistent or clearly state somewhere that you use association index as a proxy for relationship quality.

Line 157-158: Since it is a key parameter in the analysis it would be wise to explain what the associated group members are and what the others are and to put this section of the methods about how it was calculated before the condition section.

Line 181: Why did individual not receive rewards at trial 5, 10 and 15?

Line 207: Has this kit been biologically validated for use in macaques specifically? If so please cite the appropriate literature. I know that assay are really species specific when it comes to cortisol metabolites in feces (see e.g. Heistermann et al. 2006), what is the kit actually measuring in saliva? Please specify.

Line 218: Why is the unit suddenly jumping from ng/ml to ng/g? Also how was saliva concentration controlled for? (in the same way as urine concentration is controlled for using specific gravity or creatinine).

Line 222: By “Landau's linearity index” do you mean that you used the I&SI method based on matrix of winner loser interactions? Please clarify.

Line 226: why did you remove the breeding males from the hierarchy? Weren't they part of the study? Why were other males kept in the study? What defines a breeding male?

Line 227: the % of unknown relationships is really high and this is a problem even if the authors do not consider rank differences but only rank of the participant itself... Furthermore most of the

discussion about the effect of rank revolved around the idea that lower ranking individual than their partners do A and higher ranking do B whereas this is not what is tested in the models. I am really surprised that the hierarchy was not clearer since long-tailed macaques have really clear linear matrilineally based hierarchies in females.

Line 231: The authors should explain why they did not consider grooming in their index.

Line 272: Given the clear hypothesis and predictions you have I do not understand why you use an information theory approach especially for an experimental setup. This form of analysis is more suited to exploration of several parameters potentially affecting the response. E.g. a suite of ecological and anthropogenic factors potentially affecting species density or behavior. In the case of this study the parameters are clearly defined and predictions are drawn from each of them so a cohesive approach building a full model (with interaction between test condition and association index) appears more suited and more parsimonious. If the authors want to keep the analysis as is they should at least explain why they went for this approach over the null hypothesis testing approach.

I noticed that none of your model comprises random slopes. Random slopes are really important to be incorporated into model since the lack of random slopes strongly inflates type 1 error rate (see e.g. Barr et al. 2013). I would strongly advise for the inclusion of random slopes which also do not affect directly the degree of freedom (i.e. complexity) of the model and should not therefore be a limitation due to sample size.

As mentioned above, to truly test if the effect of association index on cortisol reduction is truly occurring only during cooperation (or more so during this context than during the control conditions) the authors need to build a full model comprising the interaction term between test condition and association index. This is feasible even within the AIC model selection framework.

Line 282: you cannot say that you don't use the null hypothesis testing framework and then say that you don't support the null hypothesis.

Results:

The general result section refers to table entirely for the numerical values of the results which I would have incorporated to a larger extent directly in the text to understand what the result mean. Also actual figures like "40% of high ranking individuals did X" would help grasping the effect size of the differences.

Line 292: why is it "conversely" if it is "also not"?

Line 296-298: Fig 2 does not depict the results on the rank effect.

Discussion

In line with my comments on the methods, in the discussion the authors make some claims which are not directly supported by their analysis. Overall I found the discussion relatively shallow when it could have more impact while broadening the literature survey (which is not a limited factor since the number of citations is not limited in royal society open science).

Line 317: add "with" after "contrasting"

Line 321: There is a whole literature on the effect of dominance rank and social interactions (e.g. social buffering, Young et al. 2014) on cortisol excretion which is ignored in the discussion.

Line 322: If you mention "dominant" this refers to the relationship between 2 individuals. If you mean high ranking individuals this is different since it refers to the entire hierarchy. What I mean is that in the test if you pair alpha and beta females they are both high ranking categorically but the alpha female is the only dominant one of the pair. Please rephrase.

Line 326: to know if the pattern is truly different in the social control and cooperation condition you need an interaction term in your full model. (see comment above). If the interaction term is not within your best models then the effect is not condition specific.

Line 328: you did not define "close social relationships" (see comments above).

Line 332: This is not what the citation [51] found. In their study Wittig et al. show that cortisol is decreased when cooperating, grooming or sitting in proximity to bond partners to the same extent (see the non-significant interaction between relationship quality and event in their "main

effect model". I also wonder why this paper is not mentioned in the introduction of the paper. It would give some scientific rationale to the main predictions of the paper.

Line 346: here cite for grooming in chimpanzees Crockford et al. 2013.

Line 354: This entire paragraph speculate about the different effect for the higher and the lower ranking individual of the pair in the test. Yet this is not tested here, all what is tested is the rank of the participant. As mentioned above two low ranking individuals still have a rank difference and one will be dominant over the other one, the same apply to pairs of high ranking individuals so that overall high and low ranking (to the exception of the alpha and lowest ranking individuals) can each be both dominant and subordinate in the pair depending who they are paired with. The authors should therefore refrain from drawing conclusion on the rank relationship of the pair when they don't test it.

Line 364: it is good to acknowledge the limit of your study but please expand on what it means for your overall conclusions.

Additional comment on the reply to reviewers:

Regarding the choice of a beta model: the author's choice is justified not only by the distribution of the data but also because proportion are by nature bound between 0 and 1 and poisson distribution model open end distribution. Just for that reason beta distribution is better suited. This should be clearly stated in the manuscript.

Bibliography

Barr, D. J., Levy, R., Scheepers, C. & Tily, H. J. 2013. Random effects structure for confirmatory hypothesis testing: Keep it maximal. *Journal of Memory and Language*, 68, 255-278.

Crockford, C., Wittig, R. M., Langergraber, K., Ziegler, T. E., Zuberbuehler, K. & Deschner, T. 2013. Urinary oxytocin and social bonding in related and unrelated wild chimpanzees. *Proceedings of the Royal Society B-Biological Sciences*, 280, 20122765.

Heistermann, M., Palme, R. & Ganswindt, A. 2006. Comparison of different enzymeimmunoassays for assessment of adrenocortical activity in primates based on fecal analysis. *American Journal of Primatology*, 68, 257-273.

Samuni, L., Preis, A., Mundry, R., Deschner, T., Crockford, C. & Wittig, R. M. 2017. Oxytocin reactivity during intergroup conflict in wild chimpanzees. *Proceedings of the National Academy of Sciences*, 114, 268-273.

Samuni, L., Preis, A., Deschner, T., Wittig, R. M. & Crockford, C. 2019. Cortisol and oxytocin show independent activity during chimpanzee intergroup conflict. *Psychoneuroendocrinology*, 104, 165-173.

Young, C., Majolo, B., Heistermann, M., Schülke, O. & Ostner, J. 2014. Responses to social and environmental stress are attenuated by strong male bonds in wild macaques. *Proceedings of the National Academy of Sciences*, 111, 18195-18200.

Reviewer: 2

I appreciate the authors' detailed responses to the previous comments. The revisions that were made have significantly strengthened the paper.

The one concern I still have is examining whether kinship or the presence of an infant affected cooperative success (model Q1). Both of these factors have significant biological (kin selection theory) and practical (the physical constraints of pulling in a cooperative task while holding an infant) relevance to cooperative success alone. These factors may have been more important in determining cooperative success than the IVs the authors chose to include in Model Q1 (e.g. cortisol before, social association, rank).

Minor comments:

Line 203: It is not sufficient to report rpms without describing the make and model of the centrifuge and model of the rotor. Generally speaking, the radius of rotors in different centrifuges

are different. Alternatively, you can report the g/RCF (gravity/relative centrifugal force) since it is a function of the speed that the centrifuge spins and the radius of the rotor.

ESM 2, 2nd paragraph, 2nd sentence: Should "2,5" be "2.5 min"?

ESM 2, 2nd paragraph, 3rd sentence: "...after collection of the first sample and a to the subject..."
Please rephrase.

ESM 2, 2nd paragraph, 4th sentence: Please provide a reference for this.

ESM 2, Reference #4: Use sentence case for the title of the article.

Author's Response to Decision Letter for (RSOS-191056.R0)

See Appendix A.

RSOS-191056.R1 (Revision)

Review form: Reviewer 1 (Cedric Girard-Buttoz)

Is the manuscript scientifically sound in its present form?

Yes

Are the interpretations and conclusions justified by the results?

Yes

Is the language acceptable?

Yes

Do you have any ethical concerns with this paper?

No

Have you any concerns about statistical analyses in this paper?

No

Recommendation?

Accept with minor revision (please list in comments)

Comments to the Author(s)

The authors underwent a very thorough revision of their manuscript, improving upon the major critics. The current version of the manuscript is suitable for publication following some minor changes indicated below. In particular, the authors should check that they are consistent with the terminology regarding social bonds throughout the manuscript (some "social associate" terms still remain from the previous versions).

Introduction

Line 109: Please be consistent and replace "degrees of association" by "social bond strength" since you now use the term "social bond" in the rest of the manuscript.

Results:

Line 321-323: please clarify how you come to the conclusion that "Whereas cooperation with closely bonded individuals led to a cortisol decrease, changes in cortisol were not affected by

cooperation in general (irrespective of the bond with the partner), the mere presence of a conspecific, or the pulling task itself (P3)". Is it because "condition" is not included in your set of best models? It would be useful to clarify this aspect overall for the rest of the results.

Discussion:

Line 343-346: this sentence is relatively complex to follow, please break it down into two sentences.

Line 364: please replace "close associates" by "socially bonded individuals". Please make sure to be consistent with the terminology throughout the manuscript.

Discussion:

Line 421: Replace "of" by "between".

Table 3: I do not understand what is tested in model Q3. Maybe the authors omitted something but shouldn't there be indicated here the estimates for the effect of condition (to show that cortisol response is due to cooperation and not found in the control conditions?). Please clarify this part of the results.

Decision letter (RSOS-191056.R1)

03-Feb-2020

Dear Professor Stocker:

On behalf of the Editors, I am pleased to inform you that your Manuscript RSOS-191056.R1 entitled "Cooperation with closely bonded individuals reduces cortisol levels in long-tailed macaques" has been accepted for publication in Royal Society Open Science subject to minor revision in accordance with the referee suggestions. Please find the referees' comments at the end of this email.

The reviewers and Subject Editor have recommended publication, but also suggest some minor revisions to your manuscript. Therefore, I invite you to respond to the comments and revise your manuscript.

- Ethics statement

- Data accessibility

If you wish to submit your supporting data or code to Dryad (<http://datadryad.org/>), or modify your current submission to dryad, please use the following link:
<http://datadryad.org/submit?journalID=RSOS&manu=RSOS-191056.R1>

- **Competing interests**

- **Authors' contributions**

- **Acknowledgements**

- **Funding statement**

Because the schedule for publication is very tight, it is a condition of publication that you submit the revised version of your manuscript before 12-Feb-2020. Please note that the revision deadline will expire at 00.00am on this date. If you do not think you will be able to meet this date please let me know immediately.

on behalf of Dr Mark Walton (Associate Editor) and Kevin Padian (Subject Editor)
openscience@royalsociety.org

Associate Editor Comments to Author (Dr Mark Walton):

The referee stated that you have provided a thorough revision of your manuscript, which would be suitable for publication following some minor changes. Please ensure that you fully address the referee's comments in your revised manuscript upon resubmission.

Reviewer comments to Author:
Reviewer: 1

Comments to the Author(s)

The authors underwent a very thorough revision of their manuscript, improving upon the major critics. The current version of the manuscript is suitable for publication following some minor changes indicated below. In particular, the authors should check that they are consistent with the terminology regarding social bonds throughout the manuscript (some "social associate" terms still remain from the previous versions).

Introduction

Line 109: Please be consistent and replace “degrees of association” by “social bond strength” since you now use the term “social bond” in the rest of the manuscript.

Results:

Line 321-323: please clarify how you come to the conclusion that “Whereas cooperation with closely bonded individuals led to a cortisol decrease, changes in cortisol were not affected by cooperation in general (irrespective of the bond with the partner), the mere presence of a conspecific, or the pulling task itself (P3)”. Is it because “condition” is not included in your set of best models? It would be useful to clarify this aspect overall for the rest of the results.

Discussion:

Line 343-346: this sentence is relatively complex to follow, please break it down into two sentences.

Line 364: please replace “close associates” by “socially bonded individuals”. Please make sure to be consistent with the terminology throughout the manuscript.

Discussion:

Line 421: Replace “of” by “between”.

Table 3: I do not understand what is tested in model Q3. Maybe the authors omitted something but shouldn't there be indicated here the estimates for the effect of condition (to show that cortisol response is due to cooperation and not found in the control conditions?). Please clarify this part of the results.

Author's Response to Decision Letter for (RSOS-191056.R1)

See Appendix B.

RSOS-191056.R2 (Revision)

Review form: Reviewer 1 (Cedric Girard-Buttoz)

Is the manuscript scientifically sound in its present form?

Yes

Are the interpretations and conclusions justified by the results?

Yes

Is the language acceptable?

Yes

Do you have any ethical concerns with this paper?

No

Have you any concerns about statistical analyses in this paper?

No

Recommendation?

Accept as is

Comments to the Author(s)

The authors have appropriately addressed all of my final recommendations and the manuscript is acceptable for publication as is.

Decision letter (RSOS-191056.R2)

14-Apr-2020

Dear Professor Stocker,

It is a pleasure to accept your manuscript entitled "Cooperation with closely bonded individuals reduces cortisol levels in long-tailed macaques" in its current form for publication in Royal Society Open Science.

on behalf of Dr Mark Walton (Associate Editor) and Kevin Padian (Subject Editor)
openscience@royalsociety.org

Reviewer comments to Author:

Reviewer: 1

Comments to the Author(s)

The authors have appropriately addressed all of my final recommendations and the manuscript is acceptable for publication as is.

Appendix A

Dear Reviewers, dear Editors

Thank you for reviewing our manuscript and for your constructive critique. Please, find our responses to your comments below. The references we are referring to in our answers are listed at the end of this file. When we changed the manuscript with regard to a particular comment, we refer to the lines in the marked-up copy of the manuscript. (Please note, that the line numbers are not the same in the "no markup" version.)

In particular, you'll see that in the revised manuscript we have returned to the term "social bonds" (also in the title) instead of "associates". As suggested by reviewer 1 we have elaborated more on the link between social bonds and cooperation as well as cortisol excretion, which allows the reader to better understand our hypotheses/predictions. Also, in the discussion we have added more information on social bonds, and we have revised the discussion on rank.

Though we have tried to adapt the manuscript according to your suggestions, there are a few cases in which the (previous RSPB and current) reviewers' opinions contradict each other. This concerns the following points:

"Social bond" or "social association"

Whereas previous reviewer 2 requested "social bonds" to be named "social associations", we have now followed the suggestion of current reviewer 1, used the term "social bond" and elaborated more on that in the introduction.

Statistics: Interaction vs. splitting data

In general, everyone has a different opinion about the "right" statistical approach and applying a rather new statistical approach often leads to discussions. We were trying to explain our choices in great detail and followed some of the previous reviewers' wishes. One of them was that we ran a separate model for the cooperation condition and the social control condition, rather than analyzing data from those conditions together (including the interaction of condition and social bond) like in a previous version. Current reviewer 1, however, now wants us to do it as it was in the first version. We have now decided to maintain the analyses as we have now and as they were proposed by the previous reviewer since we agree with that reviewer's rationale for the separate analyses.

Statistics: Model Q1

There still seems to be a disagreement on which independent variables should be included in model Q1. Previous reviewer 2 (RSPB) stated that we do not have predictions for the effect of sex and the presence of an infant (factors included in version 1, see table below). Therefore, and because of the small sample size we decided to keep the model simple and only include the variables of interest (version 2) plus Individual and Partner ID as random effects. Current reviewer 2, however, wants kinship and the presence of an infant to be included (version 3). Whereas we are aware of the potential importance of several factors that could influence the response variables of the respective models, only a limited number of independent variables can be included in models based on small datasets. We have, therefore, decided not to change this model (version 2).

	Condition	N (Obs.)	Dependent v.	Independent variables					
Version 1	Cooperation	14 (29)	coop. succ.	cort. before	soc.bond	rank	sex	infant	
Version 2	Cooperation	14 (29)	coop. succ.	cort. before	soc.bond	rank	-	-	-
Version 3	Cooperation	14 (29)	coop. succ.	cort. before	soc.bond	rank		infant	kin

Despite the issues raised above, we do think that these revisions have made our manuscript stronger. We would like to thank you once again for giving us the opportunity to submit a revised manuscript and hope that it meets the required changes.

Yours sincerely,
Martina Stocker on behalf of the authors

Response to Referees

Reviewer: 1

In this interesting study the authors report one rare experimental test of the effect of collaboration on cortisol excretion. The study is ambitious and appears well controlled and the results are clear cut. I am however not sure about certain of the analysis and the methods still lack certain level of details (despite the revisions). **The author should also clearly acknowledge that their definition of cooperation is not universal and that what they describe here is more commonly termed collaboration** (acting together to achieve a common goal that one cannot achieve alone) whereas cooperation in the broadest sense (in the Darwinian sense) is defined as a service provided by an individual A to another individual B at a cost for A and which benefits B.

We respectfully disagree with the reviewer here. There unfortunately still is quite some discussion about the terminology regarding cooperation, but much of that discussion is about whether cooperation is used as an overarching term that describes both (reciprocal) interactions and interactions that yield benefit for both participants immediately (sometimes referred to as mutualism) or whether it only describes the latter (see Noë, 2006). Either way, when considering our experiment, cooperation would fit to describe it. Moreover, also from a semantic point of view considering our experiment, where two individuals simultaneously have to operate an apparatus, *co-operating* seems the right term. Anyway, we have clearly mentioned in the introduction which specific definition of cooperation we are using; i.e. we are using Brosnan & de Waal's (2002) definition of the term cooperation.

In this sense behaviors such as **grooming or alarm calling** have been frequently given as example of cooperative actions. By taking a broader definition of cooperation the authors could **incorporate the literature on physiological consequences of grooming interaction for example on cortisol excretion (and on other hormones such as oxytocin) into their introduction and broaden the scope of their paper.** Even sticking strictly to their definition, the authors omit in their introduction some recent literature

on physiological correlates of collective collaborative action such as border patrol and hunting in chimpanzees (e.g. Samuni et al. 2017, 2019).

We do agree that broadening the scope of our paper by including more literature regarding cooperation in its broader definition would strengthen our paper and have now included the suggested references, though some of them only in the discussion.

Introduction:

As mentioned above, the introduction would benefit from incorporating broader literature about the **link between cooperation (in the road sense of the term) and mammalian physiology**. This would also shed light on the theoretical rationale of the predictions which partly appear not really justified by the introduction itself as it stands.

Thank you for bringing this to our attention. We have now extended our introduction and discussion and hope to have provided the reader with the appropriate background.

Line 58: replace "of" by "between"

We have now replaced it.

Line 65: replace "since" by "while".

"they only do so" refers to "show prosocial behavior" in the sentence before. Replacing "since" with "while" would, therefore, not make sense in our opinion.

Line 81: please define **degree of association** clearly in the introduction. It is important to understand your hypothesis. Also you can refer to literature showing that in societies where individuals form social bonds with each other (e.g. macaques, baboons, chimpanzees) social proximity or association in party often reflects social bond strength assessed with more direct/intimate measures such as grooming frequency. That would allow you to present your predictions and discuss your results in a social bond framework. I know that previous reviewers criticized that you used the term **social bond**, I would clearly state that you use association measures as a proxy for social bond strength and that this proxy is likely to be a good one since in other species of macaques (e.g. barbary or Assamese macaques) it was found that proximity strongly correlates with grooming time and overall social bond strength (see the work from Ostner's and Schulke's research group).

Despite the criticism of previous reviewers regarding our initial submission to RSPB, we do agree with you and have now returned to the term "social bond". We also elaborate more on that in the introduction (line 64-81).

Line 86: The authors should not assume that the reader knows what a **loose string task** is, especially for a broad audience like royal society open science. They may want to clarify the paradigm in one short sentence and mention that this is a collaborative action where two individuals act together to access a sharable reward.

We agree and have now added a sentence like you suggested (line 110).

Line 92-96: Why are the **predictions named "Q"**? Also the **prediction Q2** is **not really well introduced** by the general introduction and comes a bit out of the blue. Using the literature on oxytocin correlates of grooming (e.g. Crockford et al. 2013) would help drawing prediction on differential physiological effect of cooperation (or social interaction) in bonded and non-bonded individuals.

We have now named the predictions P instead of Q and added a paragraph on the connection of social bonds with cooperation and with cortisol. We have not included literature on oxytocin and grooming in the introduction as we have investigated neither of them (and our macaques were not able to engage in grooming behavior during our experiment). However, we do/did discuss such literature in the discussion.

Methods

The methods are overall clear but lack certain level of details in certain section which makes it hard to grasp what the authors exactly did. I am also not convinced that the way the data were analyzed is the most parsimonious and I would strongly advice to reduce the number of analysis and incorporate interactions into full models to truly test if the effect is condition dependent.

Though it appears to be preferential to reduce the number of analyses and include interactions, our questions require a different approach as important variables are not existent in all conditions. One of our questions is whether cooperative success has an effect on cortisol changes. As cooperative success does not exist in the social control, we computed a model (Q2a) using only data from the cooperation condition.

To test whether the effect of social bond (which does not exist in the non-social control) on cortisol is only due to the presence of the partner, we initially presented a model using data from the cooperation condition and the social control together which included the interaction of condition and social bond. Two reviewers from our initial submission to RSPB (see comments and our responses to that below), however, suggested not to do that. We follow(ed) their suggestion and instead split the data (model Q1a and b).

Reviewer 2, September 2018:

It is unclear what the variable "condition" represents as the authors wrote that the Cooperation and Social control conditions "were taken together" (line 266). It is unclear why Social control tests were included in this analysis as prediction Q3 expects "more pronounced drop in cortisol would occur after cooperation with affiliates" (line 85). Please rerun the analysis or better remove it as the effect of cooperation with partners with high proximity degree was already tested in the previous model.

The variable condition does refer to the conditions, however, in Q3 only 2 of the 3 conditions (cooperation and social control) are included in the data analysis. This is because we specifically want to test whether the effect of social bond during cooperation is not due to the mere presence of that affiliate. In Q2 we wanted to investigate the effect of social bonds and cooperative success. Since, however, in the social control cooperative success does not exist, we needed to create two separate models.

We do understand that without having seen table 2, in which we listed the conditions included in each model, it is difficult to follow. Therefore, we tried to draw the reader's attention to that table (I

293), and we changed “were taken together” into “when analyzing data from both social conditions (cooperation, social control)” (l 362).

Reviewer 1, April 2019:

Why did the authors chose to analyse the social conditions (cooperation, social control) together when making inferences about whether social bond or rank predicted changes in cortisol levels rather than analysing these factors (i.e. social bond and rank) in the social control condition alone? Effects could have been driven by one of the conditions rather than the combination of the two.

Thank you for your advice. To determine whether the effect of social bond (now association) was driven by the conditions we initially included an interaction effect of condition and social bond (not however rank) in our former model Q3. We do, however agree, that analyzing the social conditions (cooperation and social control) separately is a good way of investigating possible effect differences between the conditions. In the current version of our manuscript we, therefore, changed that in the respective model and also adjusted the numbers of the full models as follows:

Model Q2 a (instead of simply Q2) – cooperation data
Model Q2 b (instead of Q3) – social control data
Model Q3 (former Q4) – data from all conditions

Line 155: you call it in the introduction association index. Now it's “**quality of the relationship**”. Be consistent or clearly state somewhere that you use association index as a proxy for relationship quality.

As mentioned above, we are now mainly using the term “social bond”.

Line 157-158: Since it is a key parameter in the analysis it would be wise to explain what the associated group members are and what the others are and to put this **section of the methods** about how it was calculated before the condition section.

We have now placed the section on social association and dominance hierarchy after “Subjects and housing” and called it “Social measures” instead of “Additional measures”.

Line 181: Why did individual not receive rewards at **trial 5, 10 and 15**?

We wanted to keep the amount of food the animals would get in the cooperation condition and the social control condition as constant as possible, because getting food as such might have an effect on cortisol levels. Since we assumed that the subjects would not succeed in all 15 trials we chose to simulate 12 successful cooperation trials. Considering that the median of successful trials was 12 we feel that we made the right choice. We now also mention this in the methods section and report the median in the test sessions and the similarity with our simulation in the discussion.

Line 207: Has this kit been **biologically validated** for use in macaques specifically? If so please cite the appropriate literature. I know that assay are really species specific when it comes to **cortisol metabolites** in feces (see e.g. Heistermann et al. 2006), what is the kit actually measuring in saliva? Please specify.

Validations are indeed very important. To our knowledge the IBL salivary cortisol kit has not previously been used in macaques. Before conducting our study, we, therefore, started a biological validation process (with

very high temporal resolution) with four long-tailed macaques, of which one finished the procedure (non-invasive, no ACTH challenge). We presented this in *ESM 2 - Salivary Cortisol*.

The determination of cortisol concentrations in saliva is generally well established and the IBL cortisol saliva ELISA kit for humans was used for cortisol analyses in several other animal studies (e.g. pig saliva (Thomsson et al., 2014); cow saliva and plasma (Dzviti et al., 2019); mice serum (Meynaghizadeh-Zargar et al., 2019); rat serum (Salehpour et al., 2017)). Petow and Lewald (2012) were even able to measure non-metabolized cortisol in cattle feces with this kit. This indicates that the IBL kit can be used for a much broader purpose than only for human saliva. It is, therefore, also highly unlikely that when applied to saliva of an old-world monkey, who is comparably closely related to humans, the IBL kit is not measuring cortisol.

For more information on salivary cortisol analyses in non-human primates see (Laudenslager et al., 2006) and (Behringer and Deschner, 2017).

Possible cross reactions (see below) are reported in the assay description.

Substance	Cross Reactivity (%)	
Prednisolone	30	Cross-reactivity of other substances tested < 0.01 %
11-Desoxy-Cortisol	7.0	
Corticosterone	1.4	
Cortisone	4.2	
Prednisone	2.5	
17 α -OH-Progesterone	0.4	
Desoxy-Corticosterone	0.9	
6 α -Methyl-17 α -OH-Progesterone	0.04	

Line 218: Why is the unit suddenly jumping from **ng/ml** to **ng/g**?

Thank you for reporting that. This is a mistake that I unfortunately only discovered after we submitted the manuscript. We have now changed it to ng/ml.

Also how was **saliva concentration controlled** for? (in the same way as urine concentration is controlled for using specific gravity or creatinine).

Whereas in urine it is indeed common to control for its concentration, for salivary cortisol this is usually not done.

Behringer and colleagues (2012) found that the chewing duration and with that the amount of saliva does not affect the concentration of salivary alpha-amylase in bonobos. This indicates that there is no need to control for saliva concentration. (In the same study they also analyzed salivary cortisol without controlling for saliva concentration)

Moreover, we followed the protocol of the assay kit manufacturer IBL, which, just like e.g. the assay protocol of the company Salimetrics, does not include such steps.

Line 222: By "**Landau's linearity index**" do you mean that you used the I&SI method based on matrix of winner loser interactions? Please clarify.

Landau's linearity index is indeed based on a matrix, as is described in the accompanying reference (de Vries et al. 1993), though in this case not on winner-looser interactions but on sending and receiving of unidirectional dominance signals.

Line 226: why did you remove the **breeding males** from the hierarchy? Weren't they part of the study? Why were other males kept in the study? What defines a breeding male?

The long-tailed macaque breeding groups at the BPRC only contain one fully adult and to the group non-natal male who sires the offspring. Natal males are taken out of the group just before they become sexually mature around the age of four to prevent inbreeding, simulating natural male dispersal (van Noordwijk and van Schaik, 1985). Since the breeding male usually stands above the matrilineal hierarchy, we initially excluded him from our hierarchy calculations. Moreover, the breeding male of our study group did not meet the test criteria. During training (independent of our study) he did not focus on the task, but constantly kept an eye on his group and showed signs of anxiety. Therefore, we could not have kept him in the training area for a whole test session. Nevertheless, we have now included the breeding male into the hierarchy and updated all relevant parts in the manuscript and ESM. Our results concerning rank remain the same (except for intercepts of the full models) because all individuals that participated in our study stand below the breeding male and, therefore, they all moved one rank down. Moreover, rank in our models is standardized.

Line 227: the % of **unknown relationships** is really high and this is a problem even if the authors do not consider rank differences but only rank of the participant itself.. Furthermore most of the **discussion** about the effect of rank revolved around the idea that lower ranking individual than their partners do A and higher ranking do B whereas this is not what is tested in the models. I am really surprised that the hierarchy was not clearer since long-tailed macaques have really clear linear matrilineally based hierarchies in females.

Though the high number of unknown relationships might indeed be considered a weakness of our study, we do think that our rank calculations are a reliable estimate of about where the individual is positioned in the rank hierarchy. Whereas potential inaccuracies of rank position have major consequences for rank differences/relationship between individuals (dominant vs. subdominant), they should be less impactful when it comes to assessing if an individual is high or low ranking. Moreover, all relationships that are known are one-way relationships, indicating that they are linear. Interestingly, out of 9 mother-offspring relationships dominant behavior was only recorded in the case of the alpha-female and her two kids.

Concerning the discussion see response to comment in discussion section.

Line 231: The authors should explain why they did not consider grooming in their index.

For the calculation of the social associations we pooled incidents of grooming and being within reach of one arm length. We have now specified that in line 167.

Line 272: Given the clear hypothesis and predictions you have I do not understand why you use a **information theory approach** especially for an experimental setup. This form of analysis is more suited to exploration of several parameters potentially affecting the response. E.g. a suit of ecological and anthropogenic factors potentially affecting species density or behavior. In the case of this study the parameters are clearly defined and predictions are drawn from each of them so a cohesive approach building a full model (with interaction between test condition and association index) appears more suited and more parsimonious. If the authors want to keep the analysis as is they should at least **explain why they went for this approach over the null hypothesis testing approach**.

The IT approach is indeed regarded as particularly suitable for observational studies, but see Mundry (2011): "Occasionally, it has been suggested that the IT-based approach might be preferable in case of observational studies whereas NHST is the better option in case of an experimental study (Burnham and Anderson 2002; see also Stephens et al. 2007). However, I personally do not find this distinction to be very helpful because, on the one hand, observational studies can be 'quasi-experimental' and, on the other hand, experimental studies can be confounded by a whole set of rather uncontrollable confounding variables turning them, in fact, into 'quasi-experimental' studies." Further he says "An experimental study, on the other hand, may become rather 'observational', for instance, when the number of individuals is limited (and hence one has to take the individuals available), and some potentially confounding variables like age, sex, litter or prior testing experience should be incorporated into the statistical model. Hence, it seems that a decision between one or the other approach is best driven by the specific question addressed rather than by the somewhat arbitrary distinction between experimental and observational studies"

Since the number of our subjects is limited and we have confounding variables we prefer to use the IT approach. We have now added that in line 364.

I noticed that none of your model comprises **random slopes**. Random slopes are really important to be incorporated into model since the lack of random slopes strongly inflate type 1 error rate (see e.g. Barr et al. 2013). I would strongly advice for the inclusion of random slopes which also do not affect directly the degree of freedom (i.e. complexity) of the model and should not therefore be a limitation due to sample size.

Using a function created by Roger Mundry, we have now tested if random slopes are needed. Results indicate that only in model Q3 Condition within Individual should be included. However, when incorporating the random slopes several models (from the dredge function) failed to converge and the AIC of the full model got slightly higher (delta AIC 1.5). Therefore, the model with random slopes did not better explain the data. Following the concept of parsimony, we suggest not to use the random slopes here.

As mentioned above, to truly test if the effect of association index on cortisol reduction is truly occurring only during cooperation (or more so during this context than during the control conditions) the authors need to build a full model comprising the **interaction term** between test condition and association index. This is feasible even within the AIC model selection framework.

See answer above.

Line 282: you cannot say that you don't use the **null hypothesis testing** framework and then say that you don't support the null hypothesis.

Though we are not following the approach in which the full model is compared to the null model, we do consider the null model to be a valuable indicator of the (non-)importance of our test variables. We have now deleted "(in contrast to null hypothesis significance testing)" in our manuscript (line 364).

Results:

The general result section refers to table entirely for the numerical values of the results which I would have incorporated to a larger extent directly in the text to understand what the result mean. Also actual **figures like "40% of high ranking individuals did X"** would help grasping the effect size of the differences.

We respectfully disagree. We feel that the way we present the data here provide a better/easier read. Note that when published this text will be alongside the table (rather than at the end of the ms now) and the accompanying graphs, and we thus also provide that information immediately, and crucially, also in relation to the other results.

Line 292: why is it "conversely" if it is "also not"?

With "conversely" we were referring to the fact that in our study cortisol does not influence cooperative success and vice versa. We have now deleted "conversely".

Line 296-298: **Fig 2** does not depict the results on the rank effect.

You are right. Fig 2 only depicts the result on social association. We have now adjusted the references in the text accordingly.

Discussion

In line with my comments on the methods, in the discussion the authors make some claims which are not directly supported by their analysis. Overall I found the discussion relatively shallow when it could have more impact while broadening the literature survey (which is not a limited factor since the number of citations is not limited in royal society open science).

Line 317: add "with" after "contrasting"

We have done so now! Thank you for bringing this to our attention.

Line 321: There is a whole literature on the effect of **dominance rank** and **social interactions** (e.g. social buffering, Young et al. 2014) on **cortisol** excretion which is ignored in the discussion.

Though we are aware of the fact that there is a whole literature on the effect of dominance rank on cortisol level, we did not go too much into detail as most of these studies focus on "baseline" levels and not on short term changes of cortisol. We have, however, adjusted the respective part of

the discussion and added more information on dominance rank (line 490-498) as well as on social buffering (line 448-489).

Line 322: If you mention "**dominant**" this refers to the relationship between 2 individuals. If you mean high ranking individuals this is different since it refers to the entire hierarchy. What I mean is that in the test if you pair alpha and beta females they are both high ranking categorically but the alpha female is the only dominant one of the pair. Please rephrase.

This is indeed an important difference. To clarify this, we have now rephrased this part (line 428).

Line 326: to know if the pattern is truly different in the social control and cooperation condition you need an **interaction term** in your full model. (see comment above). If the interaction term is not within your best models then the effect is not condition specific.

See answer above.

Line 328: you did not define "**close social relationships**" (see comments above).

See answer above.

Line 332: This is not what the **citation** [51] found. In their study **Wittig** et al. show that cortisol is decreased when cooperating, grooming or sitting in proximity to bond partners to the same extent (see the non-significant interaction between relationship quality and event in their "main effect model". I also wonder why this paper is not mentioned in the introduction of the paper. It would give some scientific rationale to the main predictions of the paper.

In Wittig et al. (2016) it says: "... bond partners also downregulate HPA axis activity even during an everyday affiliation, grooming and with a weaker effect, by their mere presence when resting...". This is what we were referring to when we wrote "... close social bonds have a downregulating effect on the HPA axis, particularly during cooperation and less when only resting next to each other [51]." We do, however, understand that this could be misunderstood and have, therefore, now revised that part in the discussion (line 452). Moreover, we also included the reference in the introduction (line 77 and 81).

Line 346: here cite for grooming in chimpanzees **Crockford et al. 2013**.

With this we were actually referring to Wittig et al. (2016), who "investigated the magnitude of the difference in urinary oxytocin levels after food-sharing events compared with grooming events". However, we have now also added your suggested reference (line 478).

Line 354: This entire paragraph speculate about the different effect for the **higher and the lower ranking individual** of the pair in the test. Yet this is not tested here, all what is tested is the rank of the participant. As mentioned above two low ranking individuals still have a rank difference and one will be dominant over the other one, the same apply to pairs of high ranking individuals so that overall high and low ranking (to the

exception of the alpha and lowest ranking individuals) can each be both dominant and subordinate in the pair depending who they are paired with. The authors should therefore **refrain from drawing conclusion on the rank relationship of the pair** when they don't test it.

Line 364: it is good to acknowledge the **limit of your study** but please expand on what it means for your overall conclusions.

Thank you for your valuable critique on the discussion of the rank effect. We have now changed it and elaborated on the limitations regarding the unknown rank relationships in more detail (line 490-525).

Additional comment on the reply to reviewers:

Regarding the choice of a **beta model**: the author's choice is justified not only by the distribution of the data but also because proportions are by nature bounded between 0 and 1 and the binomial distribution model is open ended distribution. Just for that reason the beta distribution is better suited. This should be clearly stated in the manuscript.

Bibliography

Thank you for this comment. We have now followed your suggestion and added this information in line 304

Barr, D. J., Levy, R., Scheepers, C. & Tily, H. J. 2013. Random effects structure for confirmatory hypothesis testing: Keep it maximal. *Journal of Memory and Language*, 68, 255-278.

Crockford, C., Wittig, R. M., Langergraber, K., Ziegler, T. E., Zuberbuehler, K. & Deschner, T. 2013. Urinary oxytocin and social bonding in related and unrelated wild chimpanzees. *Proceedings of the Royal Society B-Biological Sciences*, 280, 20122765.

Heistermann, M., Palme, R. & Ganswindt, A. 2006. Comparison of different enzyme immunoassays for assessment of adrenocortical activity in primates based on fecal analysis. *American Journal of Primatology*, 68, 257-273.

Samuni, L., Preis, A., Mundry, R., Deschner, T., Crockford, C. & Wittig, R. M. 2017. Oxytocin reactivity during intergroup conflict in wild chimpanzees. *Proceedings of the National Academy of Sciences*, 114, 268-273.

Samuni, L., Preis, A., Deschner, T., Wittig, R. M. & Crockford, C. 2019. Cortisol and oxytocin show independent activity during chimpanzee intergroup conflict. *Psychoneuroendocrinology*, 104, 165-173.

Young, C., Majolo, B., Heistermann, M., Schülke, O. & Ostner, J. 2014. Responses to social and environmental stress are attenuated by strong male bonds in wild macaques. *Proceedings of the National Academy of Sciences*, 111, 18195-18200.

Reviewer: 2

I appreciate the authors' detailed responses to the previous comments. The revisions that were made have significantly strengthened the paper.

Thank you!

The one concern I still have is examining whether **kinship or the presence of an infant affected cooperative success (model Q1)**. Both of these factors have significant biological (kin selection theory) and practical (the physical constraints of pulling in a cooperative task while holding an infant) relevance to cooperative success alone. These factors may have been more important in determining cooperative success than the IVs the authors chose to include in Model Q1 (e.g. cortisol before, social association, rank).

We agree with you that also other factors may have been important in determining cooperative success and did in fact include at least the presence of an infant and also sex in model Q1 in our first version (see table below). Back then we only included individual as random effect.

Previous reviewer 2 of our initial submission to RSPB, however, stated that we do not have predictions for the effect of sex and the presence of an infant and that we need to control for the multiple testing of the partner. Therefore, we decided to only include the variables of interest (version 2) and add partner as random effect.

Nevertheless, we are of course interested in what could have affected cooperative success (also one of the reasons why we used the information theoretic approach). Therefore, we have now tried to add infant and kin (version 3), but since we now have two random effects and (too) many independent variables, many models (created by the dredge function) do not converge anymore i.e. the results are not reliable. For us this means that we have to choose between a model with the variables of interest plus the random effects individual and partner and a model with two more/other independent variables that is, however, missing elements needed to answer our questions.

Though other factors (kin, presence of infant, but also sex) may have been more important in determining cooperative success, we do not have enough data to reliably show that they affect cooperation (while controlling for multiple testing of the individuals) and, therefore, decide not to change that model.

	Condition	N (Obs.)	Dependent v.	Independent variables					
Version 1	Cooperation	14 (29)	coop. succ.	cort.before	soc.bond	rank	sex	infant	
Version 2	Cooperation	14 (29)	coop. succ.	cort.before	soc.bond	rank	-	-	-
Version 3	Cooperation	14 (29)	coop. succ.	cort.before	soc.bond	rank		infant	kin

Minor comments:

Line 203: It is not sufficient to report rpms without describing the make and model of the centrifuge and model of the rotor. Generally speaking, the radius of rotors in different centrifuges are different. Alternatively, you can report the **g/RCF** (gravity/relative centrifugal force) since it is a function of the speed that the centrifuge spins and the radius of the rotor.

This is indeed important. We have now added the required information on the centrifuge and the rotor (line 283).

- ESM 2, 2nd paragraph, 2nd sentence: Should "2,5" be "2.5 min"?
- ESM 2, 2nd paragraph, 3rd sentence: "...after collection of the first sample and a to the subject..." Please rephrase.
- ESM 2, 2nd paragraph, 4th sentence: Please provide a reference for this.
- ESM 2, Reference #4: Use sentence case for the title of the article.

Thank you for bringing our attention to these points. We have now adjusted
ESM 2 accordingly.

References

- Behringer, V., Deschner, T., 2017. Non-invasive monitoring of physiological markers in primates. *Horm. Behav.* doi:10.1016/j.yhbeh.2017.02.001
- Behringer, V., Deschner, T., Möstl, E., Selzer, D., Hohmann, G., 2012. Stress affects salivary alpha-Amylase activity in bonobos. *Physiol. Behav.* 105, 476-482. doi:10.1016/j.physbeh.2011.09.005
- Brosnan, S.F., de Waal, F.B.M., 2002. A proximate perspective on reciprocal altruism. *Hum. Nat.* 13, 129-152. doi:10.1007/s12110-002-1017-2
- Dzviti, M., Mapfumo, L., Muchenje, V., 2019. Relationship between saliva and blood cortisol in handled cows. *Asian-Australasian J. Anim. Sci.* 32, 734-741. doi:10.5713/ajas.18.0151
- Laudenslager, M.L., Bettinger, T., Sackett, G.P., 2006. Saliva as a Medium for Assessing Cortisol and Other Compounds in Nonhuman Primates: Collection, Assay, and Examples, in: G. Sackett & G. Ruppenthal (Eds.), *Nursery Rearing of Nonhuman Primates in the 21st Century; Developments in Primatology: Progress and Prospects*. Plenum Publishers, pp. 403-427.
- Meynaghizadeh-Zargar, R., Sadigh-Eteghad, S., Mohaddes, G., Salehpour, F., Rasta, S.H., 2019. Effects of transcranial photobiomodulation and methylene blue on biochemical and behavioral profiles in mice stress model. *Lasers Med. Sci.* doi:10.1007/s10103-019-02851-z
- Mundry, R., 2011. Issues in information theory-based statistical inference—a commentary from a frequentist's perspective. *Behav. Ecol. Sociobiol.* 65, 57-68. doi:10.1007/s00265-010-1040-y
- Noë, R., 2006. Cooperation experiments: coordination through communication versus acting apart together. *Anim. Behav.* 71, 1-18. doi:10.1016/j.anbehav.2005.03.037
- Petow, S., Lewald, J., 2012. A new quantification method for the analysis of non-metabolized faecal cortisol with a commercial ELISA kit. *Wiener Tierärztliche Monatsschrift - Vet. Med. Austria* 99.
- Salehpour, F., Rasta, S.H., Mohaddes, G., Sadigh-Eteghad, S., Salarirad, S., 2017. A comparison between antidepressant effects of transcranial near-infrared laser and citalopram in a rat model of depression, in: Madsen, S.J., Yang, V.X.D. (Eds.), . p. 100500G. doi:10.1117/12.2251598
- Thomsson, O., Ström-Holst, B., Sjunnesson, Y., Bergqvist, A.S., 2014. Validation of an enzyme-linked immunosorbent assay developed for measuring cortisol concentration in human saliva and serum for its applicability to analyze cortisol in pig saliva. *Acta Vet. Scand.* 56, 55. doi:10.1186/s13028-014-0055-1
- van Noordwijk, M.A., van Schaik, C.P., 1985. Male migration and rank acquisition in wild long-tailed macaques (*Macaca fascicularis*). *Anim. Behav.* 33, 849-861. doi:10.1016/S0003-3472(85)80019-1
- Wittig, R.M., Crockford, C., Weltring, A., Langergraber, K.E., Deschner, T., Zuberbühler, K., 2016. Social support reduces stress hormone levels in wild chimpanzees across stressful events and everyday affiliation. *Nat. Commun.* 7, 13361. doi:10.1038/ncomms13361

Appendix B

Dear Editors, dear Reviewer,

We are very happy that our manuscript RSOS-191056.R1 entitled "Cooperation with closely bonded individuals reduces cortisol levels in long-tailed macaques" has been accepted for publication in Royal Society Open Science. Thank you very much!

We particularly want to thank the reviewer for helping us to improve our manuscript. Please, find our answers to your comments below. When we changed the manuscript with regard to a particular comment, we refer to the lines in the marked-up copy of the manuscript. (Please note, that the line numbers are not the same in the "no markup" version.) We are confident that these changes will meet the final requirements for publication.

Yours sincerely,

Martina Stocker on behalf of the authors

Reviewer comments to Author:

Reviewer: 1

Comments to the Author(s)

The authors underwent a very thorough revision of their manuscript, improving upon the major critics. The current version of the manuscript is suitable for publication following some minor changes indicated below. In particular, the authors should check that they are consistent with the terminology regarding social bonds throughout the manuscript (some "social associate" terms still remain from the previous versions).

Thank you!

Introduction

Line 109: Please be consistent and replace "degrees of association" by "social bond strength" since you now use the term "social bond" in the rest of the manuscript.

Thank you for bringing that to our attention. We have now changed that throughout the whole manuscript (line 109, 158, 213, 375).

Results:

Line 321-323: please clarify how you come to the conclusion that "Whereas cooperation with closely bonded individuals led to a cortisol decrease, changes in cortisol were not affected by cooperation in general (irrespective of the bond with the partner), the mere presence of a conspecific, or the pulling task itself (P3)". Is it because "condition" is not included in your set of best models? It would be useful to clarify this aspect overall for the rest of the results.

Thanks for your advice. We have now added the following sentence to clarify this (line 327): "... This was indicated by the fact that none of

the models in the subset used for the model averaging contained the factor condition (ESM 1: Tab. S3)."
This should also explain why condition is not listed in Tab. 3

Discussion:

Line 343-346: this sentence is relatively complex to follow, please break it down into two sentences.

We have now broken it into two sentences (line 350): "Among the long-tailed macaques in this study there was no evidence for social bonds to influence cooperative behaviour. This contrasts with studies on, for example, chimpanzees [21] or the closely related, yet more egalitarian Barbary macaques [22-24], that did find a positive effect of social closeness on cooperation."

Line 364: please replace "close associates" by "socially bonded individuals". Please make sure to be consistent with the terminology throughout the manuscript.

See above.

Discussion:

Line 421: Replace "of" by "between".

Done (line 433). Thank you!

Table 3: I do not understand what is tested in model Q3. Maybe the authors omitted something but shouldn't there be indicated here the estimates for the effect of condition (to show that cortisol response is due to cooperation and not found in the control conditions?). Please clarify this part of the results.

Yes, in model Q3 we did test for the influence of the condition, but since condition did not make it into the model averaging it is not listed in this table.